# Learning with Relaxed Supervision

**Jacob Steinhardt**
Stanford University
jsteinhardt@cs.stanford.edu

**Percy Liang**
Stanford University
pliang@cs.stanford.edu

## Abstract

For weakly-supervised problems with deterministic constraints between the latent variables and observed output, learning necessitates performing inference over latent variables conditioned on the output, which can be intractable no matter how simple the model family is. Even finding a single latent variable setting that satisfies the constraints could be difficult; for instance, the observed output may be the result of a latent database query or graphics program which must be inferred. Here, the difficulty lies in not the model but the supervision, and poor approximations at this stage could lead to following the wrong learning signal entirely. In this paper, we develop a rigorous approach to relaxing the supervision, which yields asymptotically consistent parameter estimates despite altering the supervision. Our approach parameterizes a family of increasingly accurate relaxations, and jointly optimizes both the model and relaxation parameters, while formulating constraints between these parameters to ensure efficient inference. These efficiency constraints allow us to learn in otherwise intractable settings, while asymptotic consistency ensures that we always follow a valid learning signal.

## 1 Introduction

We are interested in the problem of learning from *intractable supervision*. For example, for a question answering application, we might want to learn a semantic parser that maps a question $x$ (e.g., "*Which president is from Arkansas?*") to a logical form $z$ (e.g., USPresident($e$) $\wedge$ PlaceOfBirth($e$, Arkansas)) that executes to the answer $y$ (e.g., BillClinton). If we are only given $(x, y)$ pairs as training data [1, 2, 3], then even if the model $p_\theta(z \mid x)$ is tractable, it is still intractable to incorporate the hard supervision constraint $[\mathbb{S}(z, y) = 1]$ since $z$ and $y$ live in a large space and $\mathbb{S}(z, y)$ can be complex (e.g., $\mathbb{S}(z, y) = 1$ iff $z$ executes to $y$ on a database). In addition to semantic parsing, intractable supervision also shows up in inverse graphics [4, 5, 6], relation extraction [7, 8], program induction [9], and planning tasks with complex, long-term goals [10]. As we scale to weaker supervision and richer output spaces, such intractabilities will become the norm.

One can handle the intractable constraints in various ways: by relaxing them [11], by applying them in expectation [12], or by using approximate inference [8]. However, as these constraints are part of the *supervision* rather than the *model*, altering them can fundamentally change the learning process; this raises the question of when such approximations are faithful enough to learn a good model.

In this paper, we propose a framework that addresses these questions formally, by constructing a relaxed supervision function with well-characterized statistical and computational properties. Our approach is sketched in Figure 1: we start with an intractable supervision function $q_\infty(y \mid z)$ (given by the constraint $\mathbb{S}$), together with a model family $p_\theta(z \mid x)$. We then replace $q_\infty$ by a family of functions $q_\beta(y \mid z)$ which contains $q_\infty$, giving rise to a joint model $p_{\theta,\beta}(y, z \mid x)$. We ensure tractability of inference by constraining $p_\theta(z \mid x)$ and $p_{\theta,\beta}(z \mid x, y)$ to stay close together, so that the supervision $y$ is never too surprising to the model. Finally, we optimize $\theta$ and $\beta$ subject to this tractability constraint; when $q_\beta(y \mid z)$ is properly normalized, there is always pressure to use the true

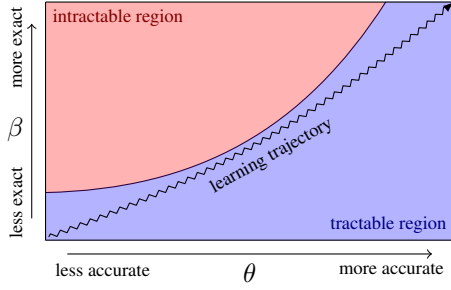

Figure 1: Sketch of our approach; we define a family of relaxations $q_\beta$ of the supervision, and then jointly optimize both $\theta$ and $\beta$. If the supervision $q_\beta$ is too harsh relative to the accuracy of the current model $p_\theta$, inference becomes intractable. In Section 4, we formulate constraints to avoid this intractable region and learn within the tractable region.

supervision $q_\infty$, and we can prove that the global optimum of $p_{\theta,\beta}$ is an asymptotically consistent estimate of the true model.

Section 2 introduces the relaxed supervision model $q_\beta(y \mid z) \propto \exp(\beta^\top \psi(z, y))$, where $\psi(z, y) = 0$ iff the constraint $\mathbb{S}(z, y)$ is satisfied (the original supervision is then obtained when $\beta = \infty$). Section 3 studies the statistical properties of this relaxation, establishing asymptotic consistency as well as characterizing the properties for any fixed $\beta$: we show roughly that both the loss and statistical efficiency degrade by a factor of $\beta_{\min}^{-1}$, the inverse of the smallest coordinate of $\beta$. In Section 4, we introduce novel *tractability constraints*, show that inference is efficient if the constraints are satisfied, and present an EM-like algorithm for constrained optimization of the likelihood. Finally, in Section 5, we explore the empirical properties of this algorithm on two illustrative examples.

## 2 Framework

We assume that we are given a partially supervised problem $x \to z \to y$ where $(x, y) \in \mathcal{X} \times \mathcal{Y}$ are observed and $z \in \mathcal{Z}$ is unobserved. We model $z$ given $x$ as an exponential family $p_\theta(z \mid x) = \exp(\theta^\top \phi(x, z) - A(\theta; x))$, and assume that $y = f(z)$ is a known deterministic function of $z$. Hence:

$$p_\theta(y \mid x) = \sum_z \mathbb{S}(z, y) \exp(\theta^\top \phi(x, z) - A(\theta; x)), \tag{1}$$

where $\mathbb{S}(z, y) \in \{0, 1\}$ encodes the constraint $[f(z) = y]$. In general, $f$ could have complicated structure, rendering inference (i.e., computing $p_\theta(z \mid x, y)$, which is needed for learning) intractable. To alleviate this, we consider projections $\pi_j$ mapping $\mathcal{Y}$ to some smaller set $\mathcal{Y}_j$; we then obtain the (hopefully simpler) constraint that $f(z)$ and $y$ match under $\pi_j$: $\mathbb{S}_j(z, y) \stackrel{\text{def}}{=} [\pi_j(f(z)) = \pi_j(y)]$. We assume $\pi_1 \times \cdots \times \pi_k$ is injective, which implies that $\mathbb{S}(z, y)$ equals the conjunction $\bigwedge_{j=1}^k \mathbb{S}_j(z, y)$. We also assume that some part of $\mathbb{S}$ (call it $\mathbb{T}(z, y)$) can be imposed tractably. We can always take $\mathbb{T} \equiv 1$, but it is better to include as much of $\mathbb{S}$ as possible because $\mathbb{T}$ will be handled exactly while $\mathbb{S}$ will be approximated. We record our assumptions below:

**Definition 2.1.** Let $\mathbb{S}(z, y)$ encode the constraint $f(z) = y$. We say that $(\mathbb{T}, \pi_1, \dots, \pi_k)$ *logically decomposes* $\mathbb{S}$ if (1) $\mathbb{S}$ implies $\mathbb{T}$ and (2) $\pi_1 \times \cdots \times \pi_k$ is injective.

Before continuing, we give three examples to illustrate the definitions above.

**Example 2.2** (Translation from unordered supervision). Suppose that given an input sentence $x$, each word is passed through the same unknown 1-to-1 substitution cipher to obtain an enciphered sentence $z$, and then ordering is removed to obtain an output $y = \text{multiset}(z)$. For example, we might have $x = abaa$, $z = dcdd$, and $y = \{c : 1, d : 3\}$. Suppose the vocabulary is $\{1, \dots, V\}$. Our constraint is $\mathbb{S}(z, y) = [y = \text{multiset}(z)]$, which logically decomposes as

$$\underbrace{[y = \overbrace{\text{multiset}(z)}^{f(z)}]}_{\mathbb{S}(z,y)} \iff \underbrace{[z_i \in y \text{ for all } i]}_{\mathbb{T}(z,y)} \wedge \bigwedge_{j=1}^V \underbrace{[\text{count}(z, j) = \overbrace{\text{count}(y, j)}^{\pi_j(y)}]}_{\mathbb{S}_j(z,y)}, \tag{2}$$

where $\text{count}(\cdot, j)$ counts the number of occurrences of the word $j$. The constraint $\mathbb{T}$ is useful because it lets us restrict attention to words in $y$ (rather than all of $\{1, \dots, V\}$), which dramatically reduces the search space. If each sentence has length $L$, then $\mathcal{Y}_j = \pi_j(\mathcal{Y}) = \{0, \dots, L\}$.

**Example 2.3** (Conjunctive semantic parsing). Suppose again that $x$ is an input sentence, and that each input word $x_i \in \{1, \dots, V\}$ maps to a predicate (set) $z_i \in \{Q_1, \dots, Q_m\}$, and the *meaning* $y$

of the sentence is the intersection of the predicates. For instance, if the sentence $x$ is *"brown dog"*, and $Q_6$ is the set of all brown objects and $Q_{11}$ is the set of all dogs, then $z_1 = Q_6$, $z_2 = Q_{11}$, and $y = Q_6 \cap Q_{11}$ is the set of all brown dogs. In general, we define $y = [\![z]\!] \stackrel{\text{def}}{=} z_1 \cap \cdots \cap z_l$. This is a simplified form of learning semantic parsers from denotations [2].

We let $\mathcal{Y}$ be every set that is obtainable as an intersection of predicates $Q$, and define $\pi_j(y) = [y \subseteq Q_j]$ for $j = 1, \ldots, m$ (so $\mathcal{Y}_j = \{0, 1\}$). Note that for all $y \in \mathcal{Y}$, we have $y = \cap_{j:\pi_j(y)=1} Q_j$, so $\pi_1 \times \cdots \times \pi_m$ is injective. We then have the following logical decomposition:

$$\underbrace{y = [\![z]\!]}_{\mathbb{S}(z,y)} \iff \underbrace{[z_i \supseteq y \text{ for all } i]}_{\mathbb{T}(z,y)} \wedge \bigwedge_{j=1}^{m} \underbrace{[\![[\![z]\!] \subseteq Q_j]\!] = \overbrace{[y \subseteq Q_j]}^{\pi_j(y)}}_{\mathbb{S}_j(z,y)}. \tag{3}$$

The first constraint $\mathbb{T}$ factors across $i$, so it can be handled tractably.

**Example 2.4** (Predicate abstraction). Next, we consider a program induction task; here the input $x$ might be "smallest square divisible by six larger than 1000", $z$ would be `argmin{i1 | mod(i1,6) = 0 and i1 = i2*i2 and i1 > 1000}`, and $y$ would be 1296; hence $\mathbb{S}(z, y) = 1$ if $z$ evaluates to $y$. Suppose that we have a collection of predicates $\pi_j$, such as $\pi_1(y) = \mod(y, 6)$, $\pi_2(y) = \mathrm{isPrime}(y)$, etc. These predicates are useful for giving partial credit; for instance, it is easier to satisfy $\mod(y, 6) = 0$ than $y = 1296$, but many programs that satisfy the former will have pieces that are also in the correct $z$. Using the $\pi_j$ to decompose $\mathbb{S}$ will therefore provide a more tractable learning signal that still yields useful information.

**Relaxing the supervision.** Returning to the general framework, let us now use $\mathbb{S}_j$ and $\mathbb{T}$ to relax $\mathbb{S}$, and thus also $p_\theta(y \mid x)$. First, define penalty features $\psi_j(z, y) = \mathbb{S}_j(z, y) - 1$, and also define $q_\beta(y \mid z) \propto \mathbb{T}(z, y) \exp\left(\beta^\top \psi(z, y)\right)$ for any vector $\beta \geq 0$. Then, $-\log q_\beta(y \mid z)$ measures how far $\mathbb{S}(z, y)$ is from being satisfied: for each violated $\mathbb{S}_j$, we incur a penalty $\beta_j$ (or infinite penalty if $\mathbb{T}$ is violated). Note that the original $q_\infty(y \mid z) = \mathbb{S}(z, y)$ corresponds to $\beta_1 = \cdots = \beta_k = +\infty$.

**Normalization constant.** The log-normalization constant $A(\beta; z)$ for $q_\beta$ is equal to $\log(\sum_{y \in \mathcal{Y}} \mathbb{T}(z, y) \exp(\beta^\top \psi(z, y)))$; this is in general difficult to compute, since $\psi$ could have arbitrary structure. Fortunately, we can uniformly upper-bound $A(\beta; z)$ by a tractable quantity $A(\beta)$:

**Proposition 2.5.** *For any $z$, we have the following bound:*

$$A(\beta; z) \leq \sum_{j=1}^{k} \log\left(1 + (|\mathcal{Y}_j| - 1) \exp(-\beta_j)\right) \stackrel{\text{def}}{=} A(\beta). \tag{4}$$

See the supplement for proof; the intuition is that, by injectivity of $\pi_1 \times \cdots \times \pi_k$, we can bound $\mathcal{Y}$ by the product set $\prod_{j=1}^{k} \mathcal{Y}_j$. We now define our joint model, which is a relaxation of (1):

$$q_\beta(y \mid z) = \mathbb{T}(z, y) \exp\left(\beta^\top \psi(z, y) - A(\beta)\right), \tag{5}$$

$$p_{\theta, \beta}(y \mid x) = \sum_z \mathbb{T}(z, y) \exp(\theta^\top \phi(x, z) + \beta^\top \psi(z, y) - A(\theta; x) - A(\beta)), \tag{6}$$

$$L(\theta, \beta) = \mathbb{E}_{x, y \sim p^*}[-\log p_{\theta, \beta}(y \mid x)], \text{ where } p^* \text{ is the true distribution.} \tag{7}$$

The relaxation parameter $\beta$ provides a trade-off between faithfulness to the original objective (large $\beta$) and tractability (small $\beta$). Importantly, $p_{\theta, \beta}(y \mid x)$ produces valid probabilities which can be meaningfully compared across different $\beta$; this will be important later in allowing us to optimize $\beta$. (Note that while $\sum_y p_{\theta, \beta}(y \mid x) < 1$ if the bound (4) is not tight, this gap vanishes as $\beta \to \infty$.)

## 3   Analysis

We now analyze the effects of relaxing supervision (i.e., taking $\beta < \infty$); proofs may be found in the supplement. We will analyze the following properties:

1. **Effect on loss**: How does the value of the relaxation parameter $\beta$ affect the (unrelaxed) loss of the learned parameters $\theta$ (assuming we had infinite data and perfect optimization)?

2. **Amount of data needed to learn**: How does $\beta$ affect the amount of data needed in order to identify the optimal parameters?

3. **Optimizing $\beta$ and consistency**: What happens if we optimize $\beta$ jointly with $\theta$? Is there natural pressure to increase $\beta$ and do we eventually recover the unrelaxed solution?

**Notation.** Let $\mathbb{E}_{p^*}$ denote the expectation under $x, y \sim p^*$, and let $L(\theta, \infty)$ denote the unrelaxed loss (see (5)–(7)). Let $L^* = \inf_\theta L(\theta, \infty)$ be the optimal unrelaxed loss and $\theta^*$ be the minimizing argument. Finally, let $\mathbb{E}_\theta$ and $\mathrm{Cov}_\theta$ denote the expectation and covariance, respectively, under $p_\theta(z \mid x)$. To simplify expressions, we will often omit the arguments from $\phi(x, z)$ and $\psi(z, y)$, and use $\mathbb{S}$ and $\neg\mathbb{S}$ for the events $[\mathbb{S}(z, y) = 1]$ and $[\mathbb{S}(z, y) = 0]$. For simplicity, assume that $\mathbb{T}(z, y) \equiv 1$.

**Effect on loss.** Suppose we set $\beta$ to some fixed value $(\beta_1, \dots, \beta_k)$ and let $\theta^*_\beta$ be the minimizer of $L(\theta, \beta)$. Since $\theta^*_\beta$ is optimized for $L(\cdot, \beta)$ rather than $L(\cdot, \infty)$, it is possible that $L(\theta^*_\beta, \infty)$ is very large; indeed, if $p_{\theta^*_\beta}(y \mid x)$ is zero for even a single outlier $(x, y)$, then $L(\theta^*_\beta, \infty)$ will be infinite. However, we can bound $\theta^*_\beta$ under an alternative loss that is less sensitive to outliers:

**Proposition 3.1.** *Let $\beta_{\min} = \min_{j=1}^k \beta_j$. Then, $\mathbb{E}_{p^*}[1 - p_{\theta^*_\beta}(y \mid x)] \leq \frac{L^*}{1 - \exp(-\beta_{\min})}$.*

The key idea in the proof is that replacing $\mathbb{S}$ with $\exp(\beta^\top \psi)$ in $p_{\theta, \beta}$ does not change the loss too much, in the sense that $\mathbb{S} \leq \exp(\beta^\top \psi) \leq \exp(-\beta_{\min}) + (1 - \exp(-\beta_{\min}))\mathbb{S}$.

When $\beta_{\min} \ll 1$, $\frac{L^*}{1 - \exp(-\beta_{\min})} \approx \frac{L^*}{\beta_{\min}}$. Hence, the error increases roughly linearly with $\beta_{\min}^{-1}$. If $\beta_{\min}$ is large and the original loss $L^*$ is small, then $L(\cdot, \beta)$ is a good surrogate. Of particular interest is the case $L^* = 0$ (perfect predictions); in this case, the relaxed loss $L(\cdot, \beta)$ also yields a perfect predictor for any $\beta > 0$. Note conversely that Proposition 3.1 is vacuous when $L^* \geq 1$.

We show in the supplement that Proposition 3.1 is essentially tight:

**Lemma 3.2.** *For any $0 < \beta_{\min} < L^*$, there exists a model with loss $L^*$ and a relaxation parameter $\beta = (\beta_{\min}, \infty, \dots, \infty)$, such that $\mathbb{E}_{p^*}[p_{\theta^*_\beta}(y \mid x)] = 0$.*

**Amount of data needed to learn.** To estimate how much data is needed to learn, we compute the *Fisher information* $\mathcal{I}_\beta \stackrel{\mathrm{def}}{=} \nabla^2_\theta L(\theta^*_\beta, \beta)$, which measures the statistical efficiency of the maximum likelihood estimator [13]. All of the equations below follow from standard properties of exponential families [14], with calculations in the supplement. For the unrelaxed loss, the Fisher information is:

$$\mathcal{I}_\infty = \mathbb{E}_{p^*}\left[\mathbb{P}_{\theta^*}[\neg\mathbb{S}]\left(\mathbb{E}_{\theta^*}[\phi \otimes \phi \mid \neg\mathbb{S}] - \mathbb{E}_{\theta^*}[\phi \otimes \phi \mid \mathbb{S}]\right)\right]. \tag{8}$$

Hence $\theta^*$ is easy to estimate if the features have high variance when $\mathbb{S} = 0$ and low variance when $\mathbb{S} = 1$. This should be true if all $z$ with $\mathbb{S}(z, y) = 1$ have similar feature values while the $z$ with $\mathbb{S}(z, y) = 0$ have varying feature values.

In the relaxed case, the Fisher information can be written to first order as

$$\mathcal{I}_\beta = \mathbb{E}_{p^*}\left[\mathrm{Cov}_{\theta^*_\beta}\left[\phi(x, z) \otimes \phi(x, z), \, -\beta^\top \psi(z, y)\right]\right] + \mathcal{O}\left(\beta^2\right). \tag{9}$$

In other words, $\mathcal{I}_\beta$, to first order, is the covariance of the penalty $-\beta^\top \psi$ with the second-order statistics of $\phi$. To interpret this, we will make the simplifying assumptions that (1) $\beta_j = \beta_{\min}$ for all $j$, and (2) the events $\neg\mathbb{S}_j$ are all disjoint. In this case, $-\beta^\top \psi = \beta_{\min}\neg\mathbb{S}$, and the covariance in (9) simplifies to

$$\mathrm{Cov}_{\theta^*_\beta}\left[\phi \otimes \phi, \, -\beta^\top \psi\right] = \beta_{\min}\mathbb{P}_{\theta^*_\beta}[\mathbb{S}]\mathbb{P}_{\theta^*_\beta}[\neg\mathbb{S}]\left(\mathbb{E}_{\theta^*_\beta}[\phi \otimes \phi \mid \neg\mathbb{S}] - \mathbb{E}_{\theta^*_\beta}[\phi \otimes \phi \mid \mathbb{S}]\right). \tag{10}$$

Relative to (8), we pick up a $\beta\mathbb{P}_{\theta^*_\beta}[\mathbb{S}]$ factor. If we further assume that $\mathbb{P}_{\theta^*_\beta}[\mathbb{S}] \approx 1$, we see that the amount of data required to learn under the relaxation increases by a factor of roughly $\beta_{\min}^{-1}$.

**Optimizing $\beta$.** We now study the effects of optimizing both $\theta$ and $\beta$ jointly. Importantly, joint optimization recovers the true distribution $p_{\theta^*}$ in the infinite data limit:

**Proposition 3.3.** *Suppose the model is well-specified: $p^*(y \mid x) = p_{\theta^*}(y \mid x)$ for all $x, y$. Then, all global optima of $L(\theta, \beta)$ satisfy $p_{\theta, \beta}(y \mid x) = p^*(y \mid x)$; one such optimum is $\theta = \theta^*$, $\beta = \infty$.*

There is thus always pressure to send $\beta$ to $\infty$ and $\theta$ to $\theta^*$. The key fact in the proof is that the log-loss $L(\theta, \beta)$ is never smaller than the conditional entropy $H_{p^*}(y \mid x)$, with equality iff $p_{\theta,\beta} = p^*$.

**Summary.** Based on our analyses above, we can conclude that relaxation has the following impact:

- **Loss:** The loss increases by a factor of $\beta_{\min}^{-1}$ in the worst case.
- **Amount of data:** In at least one regime, the amount of data needed to learn is $\beta_{\min}^{-1}$ times larger.

The general theme is that the larger $\beta$ is, the better the statistical properties of the maximum-likelihood estimator. However, larger $\beta$ also makes the distribution $p_{\theta,\beta}$ less tractable, as $q_\beta(y \mid z)$ becomes concentrated on a smaller set of $y$'s. This creates a trade-off between computational efficiency (small $\beta$) and statistical accuracy (large $\beta$). We explore this trade-off in more detail in the next section, and show that in some cases we can get the best of both worlds.

## 4 Constraints for Efficient Inference

In light of the previous section, we would like to make $\beta$ as large as possible; on the other hand, if $\beta$ is too large, we are back to imposing $\mathbb{S}$ exactly and inference becomes intractable. We would therefore like to optimize $\beta$ subject to a *tractability constraint* ensuring that we can still perform efficient inference, as sketched earlier in Figure 1. We will use rejection sampling as the inference procedure, with the acceptance rate as a measure of tractability.

To formalize our approach, we assume that the model $p_\theta(z \mid x)$ and the constraint $\mathbb{T}(z, y)$ are jointly tractable, so that we can efficiently draw exact samples from

$$p_{\theta,\mathbb{T}}(z \mid x, y) \stackrel{\text{def}}{=} \mathbb{T}(z, y) \exp\left(\theta^\top \phi(x, z) - A_{\mathbb{T}}(\theta; x, y)\right), \tag{11}$$

where $A_{\mathbb{T}}(\theta; x, y) = \log(\sum_z \mathbb{T}(z, y) \exp(\theta^\top \phi(x, z)))$. Most learning algorithms require the conditional expectations of $\phi$ and $\psi$ given $x$ and $y$; we therefore need to sample the distribution

$$p_{\theta,\beta}(z \mid x, y) = \mathbb{T}(z, y) \exp\left(\theta^\top \phi(x, z) + \beta^\top \psi(z, y) - A(\theta, \beta; x, y)\right), \text{ where} \tag{12}$$

$$A(\theta, \beta; x, y) \stackrel{\text{def}}{=} \log\left(\sum_z \mathbb{T}(z, y) \exp(\theta^\top \phi(x, z) + \beta^\top \psi(z, y))\right). \tag{13}$$

Since $\beta^\top \psi \le 0$, we can draw exact samples from $p_{\theta,\beta}$ using rejection sampling: (1) sample $z$ from $p_{\theta,\mathbb{T}}(\cdot \mid x, y)$, and (2) accept with probability $\exp(\beta^\top \psi(z, y))$. If the acceptance rate is high, this algorithm lets us tractably sample from (12). Intuitively, when $\theta$ is far from the optimum, the model $p_\theta$ and constraints $\mathbb{S}_j$ will clash, necessitating a small value of $\beta$ to stay tractable. As $\theta$ improves, more of the constraints $\mathbb{S}_j$ will be satisfied automatically under $p_\theta$, allowing us to increase $\beta$.

Formally, the expected number of samples is the inverse of the acceptance probability and can be expressed as (see the supplement for details)

$$\left(\sum_z p_{\theta,\mathbb{T}}(z \mid x, y) \exp(\beta^\top \psi(z, y))\right)^{-1} = \exp\left(A_{\mathbb{T}}(\theta; x, y) - A(\theta, \beta; x, y)\right). \tag{14}$$

We can then minimize the loss $L(\theta, \beta) = A(\theta; x) + A(\beta) - A(\theta, \beta; x, y)$ (see (6)–(7) and (13)) subject to the tractability constraint $\mathbb{E}_{x,y}[\exp\left(A_{\mathbb{T}}(\theta; x, y) - A(\theta, \beta; x, y)\right)] \le \tau$, where $\tau$ is our computational budget. While one might have initially worried that rejection sampling will perform poorly, this constraint *guarantees* that it will perform well by bounding the number of rejections.

**Implementation details.** To minimize $L$ subject to a constraint on (14), we will develop an EM-like algorithm; the algorithm maintains an inner approximation to the constraint set as well as an upper bound on the loss, both of which will be updated with each iteration of the algorithm. These bounds are obtained by linearizing $A(\theta, \beta; x, y)$; more precisely, for any $(\tilde{\theta}, \tilde{\beta})$ we have by convexity:

$$A(\theta, \beta; x, y) \ge \tilde{A}(\theta, \beta; x, y) \stackrel{\text{def}}{=} A(\tilde{\theta}, \tilde{\beta}; x, y) + (\theta - \tilde{\theta})^\top \tilde{\phi} + (\beta - \tilde{\beta})^\top \tilde{\psi}, \tag{15}$$

$$\text{where } \tilde{\phi} \stackrel{\text{def}}{=} \sum_z p_{\tilde{\theta},\tilde{\beta}}(z \mid x, y)\phi(x, z), \quad \tilde{\psi} \stackrel{\text{def}}{=} \sum_z p_{\tilde{\theta},\tilde{\beta}}(z \mid x, y)\psi(z, y).$$

We thus obtain a bound $\tilde{L}$ on the loss $L$, as well as a tractability constraint $\mathcal{C}_1$, which are both convex:

$$\text{minimize} \ \mathbb{E}_{p^*} \left[ A(\theta; x) + A(\beta) - \tilde{A}(\theta, \beta; x, y) \right] \quad (\tilde{L})$$

$$\text{subject to} \ \mathbb{E}_{p^*} \left[ \exp \left( A_{\mathbb{T}}(\theta; x, y) - \tilde{A}(\theta, \beta; x, y) \right) \right] \leq \tau. \quad (\mathcal{C}_1)$$

We will iteratively solve the above minimization, and then update $\tilde{L}$ and $\mathcal{C}_1$ using the minimizing $(\theta, \beta)$ from the previous step. Note that the minimization itself can be done without inference; we only need to do inference when updating $\tilde{\phi}$ and $\tilde{\psi}$. Since inference is tractable at $(\tilde{\theta}, \tilde{\beta})$ by design, we can obtain unbiased estimates of $\tilde{\phi}$ and $\tilde{\psi}$ using the rejection sampler described earlier. We can also estimate $A(\tilde{\theta}, \tilde{\beta}; x, y)$ at the same time by using samples from $p_{\tilde{\theta}, \mathbb{T}}$ and the relation (14).

A practical issue is that $\mathcal{C}_1$ becomes overly stringent when $(\theta, \beta)$ is far away from $(\tilde{\theta}, \tilde{\beta})$. It is therefore difficult to make large moves in parameter space, which is especially bad for getting started initially. We can solve this using the trivial constraint

$$\exp \left( \sum_{j=1}^{k} \beta_j \right) \leq \tau, \quad (\mathcal{C}_0)$$

which will also ensure tractability. We use $(\mathcal{C}_0)$ for several initial iterations, then optimize the rest of the way using $(\mathcal{C}_1)$. To avoid degeneracies at $\beta = 0$, we also constrain $\beta \geq \epsilon$ in all iterations. We will typically take $\epsilon = 1/k$, which is feasible for $(\mathcal{C}_0)$ assuming $\tau \geq \exp(1)$.[1]

To summarize, we have obtained an iterative algorithm for jointly minimizing $L(\theta, \beta)$, such that $p_{\theta, \beta}(z \mid x, y)$ always admits efficient rejection sampling. Pseudocode is provided in Algorithm 1; note that all population expectations $\mathbb{E}_{p^*}$ should now be replaced with sample averages.

---

**Algorithm 1** Minimizing $L(\theta, \beta)$ while guaranteeing tractable inference.

---

Input training data $(x^{(i)}, y^{(i)})_{i=1}^{n}$.
Initialize $\tilde{\theta} = 0$, $\tilde{\beta}_j = \epsilon$ for $j = 1, \ldots, k$.
**while** not converged **do**
    Estimate $\tilde{\phi}^{(i)}$, $\tilde{\psi}^{(i)}$, and $A(\tilde{\theta}, \tilde{\beta}; x^{(i)}, y^{(i)})$ for $i = 1, \ldots, n$ by sampling $p_{\tilde{\theta}, \tilde{\beta}}(z \mid x^{(i)}, y^{(i)})$.
    Estimate the functions $\tilde{A}(\theta, \beta; x^{(i)}, y^{(i)})$ using the output from the preceding step.
    Let $(\hat{\theta}, \hat{\beta})$ be the solution to

$$\underset{\theta, \beta}{\text{minimize}} \ \frac{1}{n} \sum_{i=1}^{n} \left( A(\theta; x^{(i)}) + A(\beta) - \tilde{A}(\theta, \beta; x^{(i)}, y^{(i)}) \right)$$

$$\text{subject to} \ (\mathcal{C}_0), \quad \beta_j \geq \epsilon \text{ for } j = 1, \ldots, k$$

    Update $(\tilde{\theta}, \tilde{\beta}) \leftarrow (\hat{\theta}, \hat{\beta})$.
**end while**
Repeat the same loop as above, with the constraint $(\mathcal{C}_0)$ replaced by $(\mathcal{C}_1)$.
Output $(\tilde{\theta}, \tilde{\beta})$.

---

## 5 Experiments

We now empirically explore our method's behavior. All of our code, data, and experiments may be found on the CodaLab worksheet for this paper at `https://www.codalab.org/worksheets/0xc9db508bb80446d2b66cbc8e2c74c052/`, which also contains more detailed plots beyond those shown here. We would like to answer the following questions:

- **Fixed $\beta$:** For a fixed $\beta$, how does the relaxation parameter $\beta$ affect the learned parameters? What is the trade-off between accuracy and computation as we vary $\beta$?

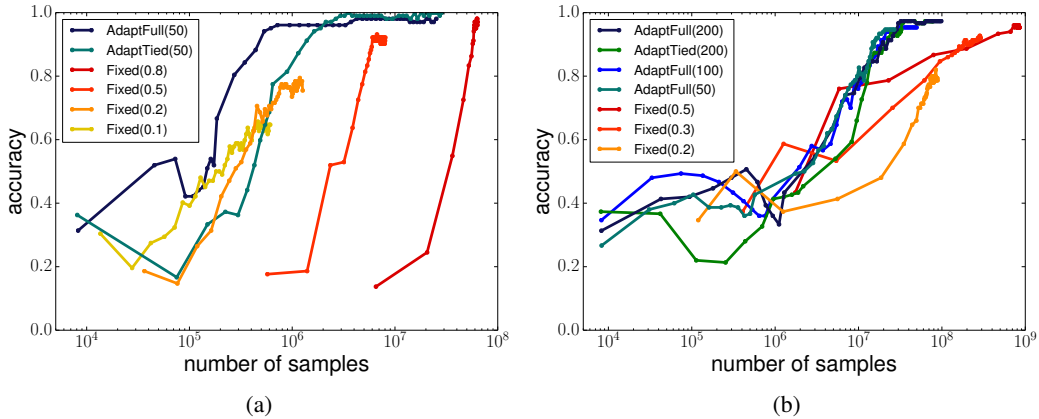

Figure 2: (a) Accuracy versus computation (measured by number of samples drawn by the rejection sampler) for the unordered translation task. (b) Corresponding plot for the conjunctive semantic parsing task. For both tasks, the FIXED method needs an order of magnitude more samples to achieve comparable accuracy to either adaptive method.

- **Adapting $\beta$:** Does optimizing $\beta$ affect performance? Is the per-coordinate adaptivity of our relaxation advantageous, or can we set all coordinates of $\beta$ to be equal? How does the computational budget $\tau$ (from $\mathcal{C}_0$ and $\mathcal{C}_1$) impact the optimization?

To answer these questions, we considered using a fixed $\beta$ (FIXED($\beta$)), optimizing $\beta$ with a computational constraint $\tau$ (ADAPTFULL($\tau$)), and performing the same optimization with all coordinates of $\beta$ constrained to be equal (ADAPTTIED($\tau$)). For optimization, we used Algorithm 1, using $S = 50$ samples to approximate each $\tilde{\phi}^{(i)}$ and $\tilde{\psi}^{(i)}$, and using the solver SNOPT [15] for the inner optimization. We ran Algorithm 1 for 50 iterations; when $\beta$ is not fixed, we apply the constraint ($\mathcal{C}_0$) for the first 10 iterations and ($\mathcal{C}_1$) for the remaining 40 iterations; when it is fixed, we do not apply any constraint.

**Unordered translation.** We first consider the translation task from Example 2.2. Recall that we are given a vocabulary $[V] \stackrel{\text{def}}{=} \{1, \ldots, V\}$, and wish to recover an unknown 1-1 substitution cipher $c : [V] \to [V]$. Given an input sentence $x_{1:L}$, the latent $z$ is the result of applying $c$, where $z_i$ is $c(x_i)$ with probability $1 - \delta$ and uniform over $[V]$ with probability $\delta$. To model this, we define a feature $\phi_{u,v}(x, z)$ that counts the number of times that $x_i = u$ and $z_i = v$; hence, $p_\theta(z \mid x) \propto \exp(\sum_{i=1}^{L} \theta_{x_i, z_i})$. Recall also that the output $y = \text{multiset}(z)$.

In our experiments, we generated $n = 100$ sentences of length $L = 20$ with vocabulary size $V = 102$. For each pair of adjacent words $(x_{2i-1}, x_{2i})$, we set $x_{2i-1} = 3j + 1$ with $j$ drawn from a power law distribution on $\{0, \ldots, V/3 - 1\}$ with exponent $r \geq 0$; we then set $x_{2i}$ to $3j + 2$ or $3j + 3$ with equal probability. This ensures that there are pairs of words that co-occur often (without which the constraint $\mathbb{T}$ would already solve the problem).

We set $r = 1.2$ and $\delta = 0.1$, which produces a moderate range of word frequencies as well as a moderate noise level (we also considered setting either $r$ or $\delta$ to 0, but omitted these results because essentially all methods achieved ceiling accuracy; the interested reader may find them in our CodaLab worksheet). We set the computational budget $\tau = 50$ for the constraints $\mathcal{C}_0$ and $\mathcal{C}_1$, and $\epsilon = \frac{1}{L}$ as the lower bound on $\beta$. To measure accuracy, we look at the fraction of words whose modal prediction under the model corresponds to the correct mapping.

We plot accuracy versus computation (i.e., cumulative number of samples drawn by the rejection sampler up through the current iteration) in Figure 2a; note that the number of samples is plotted on a log-scale. For the FIXED methods, there is a clear trade-off between computation and accuracy, with multiplicative increases in computation needed to obtain additive increases in accuracy. The adaptive methods completely surpass this trade-off curve, achieving higher accuracy than FIXED(0.8) while using an order of magnitude less computation. The ADAPTFULL and ADAPTTIED methods achieve similar results to each other; in both cases, all coordinates of $\beta$ eventually obtained their maximum value of 5.0, which we set as a cap for numerical reasons, and which corresponds closely to imposing the exact supervision signal.

**Conjunctive semantic parsing.** We also ran experiments on the semantic parsing task from Example 2.3. We used vocabulary size $V = 150$, and represented each predicate $Q$ as a subset of $[U]$, where $U = 300$. The five most common words in $[V]$ mapped to the empty predicate $Q = [U]$, and the remaining words mapped to a random subset of $85\%$ of $[U]$. We used $n = 100$ and sentence length $L = 25$. Each word in the input was drawn independently from a power law with $r = 0.8$. A word was mapped to its correct predicate with probability $1 - \delta$ and to a uniformly random predicate with probability $\delta$, with $\delta = 0.1$. We constrained the denotation $y = [\![z]\!]$ to have non-zero size by re-generating each examples until this constraint held. We used the same model $p_\theta(z \mid x)$ as before, and again measured accuracy based on the fraction of the vocabulary for which the modal prediction was correct. We set $\tau = 50, 100, 200$ to compare the effect of different computational budgets.

Results are shown in Figure 2b. Once again, the adaptive methods substantially outperform the FIXED methods. We also see that the accuracy of the algorithm is relatively invariant to the computational budget $\tau$ — indeed, for all of the adaptive methods, all coordinates of $\beta$ eventually obtained their maximum value, meaning that we were always using the exact supervision signal by the end of the optimization. These results are broadly similar to the translation task, suggesting that our method generalizes across tasks.

## 6   Related Work and Discussion

For a fixed relaxation $\beta$, our loss $L(\theta, \beta)$ is similar to the Jensen risk bound defined by Gimpel and Smith [16]. For varying $\beta$, our framework is similar in spirit to annealing, where the entire objective is relaxed by exponentiation, and the relaxation is reduced over time. An advantage of our method is that we do not have to pick a fixed annealing schedule; it falls out of learning, and moreover, each constraint can be annealed at its own pace.

Under model well-specification, optimizing the relaxed likelihood recovers the same distribution as optimizing the original likelihood. In this sense, our approach is similar in spirit to approaches such as pseudolikelihood [17, 18] and, more distantly, reward shaping in reinforcement learning [19].

There has in the past been considerable interest in specifying and learning under constraints on model predictions, leading to a family of ideas including constraint-driven learning [11], generalized expectation criteria [20, 21], Bayesian measurements [22], and posterior regularization [23]. These ideas are nicely summarized in Section 4 of [23], and involve relaxing the constraint either by using a variational approximation or by applying the constraint in expectation rather than pointwise (e.g., replacing the constraint $h(x, z, y) \geq 1$ with $\mathbb{E}[h(x, z, y)] \geq 1$). This leads to tractable inference when the function $h$ can be tractably incorporated as a factor in the model, which is the case for many problems of interest (including the translation task in this paper). In general, however, inference will be intractable even under the relaxation, or the relaxation could lead to different learned parameters; this motivates our framework, which handles a more general class of problems and has asymptotic consistency of the learned parameters.

The idea of learning with explicit constraints on computation appears in the context of prioritized search [24], MCMC [25, 26], and dynamic feature selection [27, 28, 29]. These methods focus on keeping the model tractable; in contrast, we assume a tractable model and focus on the supervision. While the parameters of the model can be informed by the supervision, relaxing the supervision as we do could fundamentally alter the learning process, and requires careful analysis to ensure that we stay grounded to the data. As an analogy, consider driving a car with a damaged steering wheel (approximate model) versus not being able to see the road (approximate supervision); intuitively, the latter appears to pose a more fundamental challenge.

Intractable supervision is a key bottleneck in many applications, and will only become more so as we incorporate more sophisticated logical constraints into our statistical models. While we have laid down a framework that grapples with this issue, there is much to be explored—e.g., deriving stochastic updates for optimization, as well as tractability constraints for more sophisticated inference methods.

**Acknowledgments.** The first author was supported by a Fannie & John Hertz Fellowship and an NSF Graduate Research Fellowship. The second author was supported by a Microsoft Research Faculty Fellowship. We are also grateful to the referees for their valuable comments.

## Footnotes

[1]If only some of the constraints $\mathbb{S}_j$ are active for each $y$ (e.g. for translation we only have to worry about the words that actually appear in the output sentence), then we need only include those $\beta_j$ in the sum for $(\mathcal{C}_0)$. This can lead to substantial gains, since now $k$ is effectively the sentence length rather than the vocabulary size.

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
