[Supplementary Material 1]

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

[2]Here we use the fact that $\mathrm{KL}\left(p \| q\right) \stackrel{\mathrm{def}}{=} \mathbb{E}_p[\log p - \log q]$ is non-negative as long as $p$ normalizes and $q$ sub-normalizes, which is true for $q = p_{\theta,\beta}$ by Proposition 2.5.

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

# A Proofs for Section 2

*Proof of Proposition 2.5.* Let $\iota \stackrel{\text{def}}{=} \pi_1 \times \cdots \times \pi_k$. Since $\iota$ is injective by assumption, we have

$$\exp(A(\beta; z)) = \sum_{y \in \mathcal{Y}} \exp(\beta^\top \psi(z, y)) \tag{16}$$

$$= \sum_{p \in \iota(\mathcal{Y})} \exp(\beta^\top \psi(z, \iota^{-1}(p))) \tag{17}$$

$$= \sum_{p \in \iota(\mathcal{Y})} \exp\left(-\sum_{j=1}^{k} \beta_j \mathbb{I}[\pi_j(f(z)) \neq p_j]\right) \tag{18}$$

$$\leq \sum_{p \in \prod_j \mathcal{Y}_j} \exp\left(-\sum_{j=1}^{k} \beta_j \mathbb{I}[\pi_j(f(z)) \neq p_j]\right) \tag{19}$$

$$= \prod_{j=1}^{k} \sum_{p_j \in \mathcal{Y}_j} \exp\left(-\beta_j \mathbb{I}[\pi_j(f(z)) \neq p_j]\right) \tag{20}$$

$$= \prod_{j=1}^{k} \left(1 + (|\mathcal{Y}_j|-1) \exp(-\beta_j)\right), \tag{21}$$

as was to be shown. $\qquad \square$

# B Proofs for Section 3

## B.1 Effect on loss

*Proof of Proposition 3.1.* Note that we have

$$L^* = \mathbb{E}_{p^*}[-\log p_{\theta^*}(y \mid x)] \tag{22}$$
$$= \mathbb{E}_{p^*}[-\log \mathbb{E}_{z \sim p_{\theta^*}}[\mathbb{S}(z, y)]] \tag{23}$$
$$\stackrel{(a)}{\geq} \mathbb{E}_{p^*}[-\log \mathbb{E}_{z \sim p_{\theta^*}}[\exp(\beta^\top \psi(z, y))]] \tag{24}$$
$$= \mathbb{E}_{p^*}[-\log \mathbb{E}_{z \sim p_{\theta^*}}[p_\beta(y \mid z)\exp(A(\beta))]] \tag{25}$$
$$= \mathbb{E}_{p^*}[-\log p_{\theta^*,\beta}(y \mid x) - A(\beta)] \tag{26}$$
$$= L(\theta^*, \beta) - A(\beta) \tag{27}$$
$$\stackrel{(b)}{\geq} L(\theta^*_\beta, \beta) - A(\beta). \tag{28}$$

Here (a) follows because $\mathbb{S}(z, y) \leq \exp(\beta^\top \psi(z, y))$, since the latter is non-negative and is 1 when $\mathbb{S}(z, y) = 1$; (b) follows because $\theta^*_\beta$ is the minimizer of $L(\cdot, \beta)$. Continuing:

$$L(\theta^*_\beta, \beta) - A(\beta) = \mathbb{E}_{p^*}[-\log \mathbb{E}_{z \sim p_{\theta^*_\beta}}[\exp(\beta^\top \psi(z, y))]] \tag{29}$$

$$\stackrel{(c)}{\geq} \mathbb{E}_{p^*}[-\log \mathbb{E}_{z \sim p_{\theta^*_\beta}}[\exp(-\beta_{\min}(1 - \mathbb{S}(z, y)))]] \tag{30}$$

$$= \mathbb{E}_{p^*}[-\log(p_{\theta^*_\beta}(y \mid x) + (1 - p_{\theta^*_\beta}(y \mid x))\exp(-\beta_{\min}))] \tag{31}$$

$$= \mathbb{E}_{p^*}[-\log(1 - (1 - p_{\theta^*_\beta}(y \mid x))(1 - \exp(-\beta_{\min})))] \tag{32}$$

$$\stackrel{(d)}{\geq} \mathbb{E}_{p^*}[(1 - p_{\theta^*_\beta}(y \mid x))(1 - \exp(-\beta_{\min}))]. \tag{33}$$

Again, (c) follows because $\beta^\top \psi(z, y) \leq -\beta_{\min}(1 - \mathbb{S}(z, y))$, and (d) follows because $-\log(1 - x) \geq x$ for $x \leq 1$. Putting these together, we have $L^* \geq (1 - \exp(-\beta_{\min}))\mathbb{E}_{p^*}[1 - p_{\theta^*_\beta}(y \mid x)]$, which yields the desired result. $\qquad \square$

*Proof of Lemma 3.2.* We will show a stronger result: any model and relaxation can be slightly modified to cause $\mathbb{E}_{p^*}[p_{\theta^*_\beta}(y \mid x)]$ to be zero, in a way that is demonstrated below (though the modified model will no longer be an exponential family).

Given any $\mathbb{S}_{1:k}$, construct a new point $z_0$ such that $\mathbb{S}_{1:k}(z_0, y) = 1$ for all $y$, and add a new constraint $\mathbb{S}_0(z, y) = [z \neq z_0]$. Then $\mathbb{S}(z_0, y) = 0$ for all $y$, so we never want to place mass on $z_0$ under the unrelaxed supervision. In addition, extend the model family to allow the single additional distribution $p'(z \mid x) = \mathbb{I}[z = z_0]$.

Now, suppose $\beta_{1:k} = \infty$ and $\beta_0 = \beta_{\min}$. Then, for any $\theta$, we have $L(\theta, \beta) = A(\beta) + L(\theta, \infty)$, since $p_\theta$ places no mass on $z_0$; therefore, $L(\theta, \beta) \geq A(\beta) + L^*$ for all $\theta$. On the other hand, we have $L(p', \beta) = A(\beta) + \beta_{\min}$. If $\beta_{\min} < L^*$, we will thus use $p'$ and shift all of the mass to $z_0$, thereby placing zero mass on the correct answer. $\qquad\square$

Note that the proof required constructing a "bad" $z_0$ that satisfied almost all the constraints for many values of $y$ at once. It seems straightforward to avoid this in practice, and so it would be interesting to find assumptions under which we obtain a better relative loss bound than Proposition 3.1.

## B.2   Amount of data needed to learn

For the next few derivations we will make extensive use of the relation $\log p_{\theta,\beta}(y \mid x) = A(\theta, \beta; x, y) - A(\theta; x)$, where $A(\theta, \beta; x, y) \overset{\text{def}}{=} \log \left( \sum_z \exp \left( \theta^\top \phi(x, z) + \beta^\top \psi(z, y) \right) \right)$. Note that the preceding definition is consistent with (13) since we assume throughout Section 3 that $\mathbb{T} \equiv 1$. We will also use the following properties of log-partition functions:

$$\nabla_\theta A(\theta, \beta; x, y) = \mathbb{E}_{z \sim p_{\theta,\beta}(\cdot \mid x, y)}[\phi(x, z)] \tag{34}$$

$$= \frac{\mathbb{E}_{z \sim p_\theta(\cdot \mid x)}[\phi(x, z) \exp(\beta^\top \psi(z, y))]}{\mathbb{E}_{z \sim p_\theta(\cdot \mid x)}[\exp(\beta^\top \psi(z, y))]}, \tag{35}$$

$$\nabla_\theta^2 A(\theta, \beta; x, y) = -(\nabla_\theta A)(\nabla_\theta A)^\top + \mathbb{E}_{z \sim p_{\theta,\beta}(\cdot \mid x, y)}[\phi(x, z) \otimes \phi(x, z)] \tag{36}$$

$$= -(\nabla_\theta A)(\nabla_\theta A)^\top + \frac{\mathbb{E}_{z \sim p_\theta(\cdot \mid x)}[(\phi(x, z) \otimes \phi(x, z)) \exp(\beta^\top \psi(z, y))]}{\mathbb{E}_{z \sim p_\theta(\cdot \mid x)}[\exp(\beta^\top \psi(z, y))]}. \tag{37}$$

Here we use $\nabla_\theta A$ as short-hand for $\nabla_\theta A(\theta, \beta; x, y)$. These $\nabla_\theta A$ terms will always cancel out in the sequel, so they can be safely ignored. (The cancellation occurs because we always end up subtracting two log-normalization constants, whose gradients must be equal by first-order optimality conditions.) Analogous properties to those above hold for $A(\theta; x)$:

$$\nabla_\theta A(\theta; x) = \mathbb{E}_{z \sim p_\theta(\cdot \mid x)}[\phi(x, z)], \tag{38}$$

$$\nabla_\theta^2 A(\theta; x) = -(\nabla_\theta A)(\nabla_\theta A)^\top + \mathbb{E}_{z \sim p_\theta(\cdot \mid x)}[\phi(x, z) \otimes \phi(x, z)]. \tag{39}$$

In this case, $\nabla_\theta A$ is short-hand for $\nabla_\theta A(\theta; x)$.

*Proof of* (8). We have

$$\mathcal{I}_\infty = \nabla_\theta^2[-\log p_{\theta^*, \infty}(y \mid x)] \tag{40}$$

$$= \nabla_\theta^2 [A(\theta^*; x) - A(\theta^*, \infty; x, y)] \tag{41}$$

$$= \mathbb{E}_{\theta^*}[\phi(x, z) \otimes \phi(x, z)] - \frac{\mathbb{E}_{\theta^*}[(\phi(x, z) \otimes \phi(x, z))\mathbb{S}(z, y)]}{\mathbb{E}_{\theta^*}[\mathbb{S}(z, y)]} \tag{42}$$

$$= \mathbb{E}_{\theta^*}[\phi(x, z) \otimes \phi(x, z)] - \mathbb{E}_{\theta^*}[\phi(x, z) \otimes \phi(x, z) \mid \mathbb{S}] \tag{43}$$

$$= (\mathbb{P}[\neg\mathbb{S}]\mathbb{E}_{\theta^*}[\phi \otimes \phi \mid \neg\mathbb{S}] + \mathbb{P}[\mathbb{S}]\mathbb{E}_{\theta^*}[\phi \otimes \phi \mid \mathbb{S}]) - \mathbb{E}_{\theta^*}[\phi \otimes \phi \mid \mathbb{S}] \tag{44}$$

$$= \mathbb{P}_{\theta^*}[\neg\mathbb{S}] (\mathbb{E}_{\theta^*}[\phi \otimes \phi \mid \neg\mathbb{S}] - \mathbb{E}_{\theta^*}[\phi \otimes \phi \mid \mathbb{S}]). \tag{45}$$

The result follows by taking expectations. $\qquad\square$

*Proof of* (9). We have

$$\mathcal{I}_\beta = \nabla_\theta^2[-\log p_{\theta_\beta^*,\beta}(y \mid x)] \tag{46}$$

$$= \nabla_\theta^2 \left[ A(\theta_\beta^*; x) - A(\theta_\beta^*, \beta; x, y) \right] \tag{47}$$

$$= \mathbb{E}_{\theta_\beta^*}[\phi(x,z) \otimes \phi(x,z)] - \frac{\mathbb{E}_{\theta_\beta^*}[(\phi(x,z) \otimes \phi(x,z)) \exp(\beta^\top \psi)]}{\mathbb{E}_{\theta_\beta^*}[\exp(\beta^\top \psi)]} \tag{48}$$

$$= \frac{\mathbb{E}_{\theta_\beta^*}[\phi \otimes \phi]\mathbb{E}_{\theta_\beta^*}[\exp(\beta^\top \psi)] - \mathbb{E}_{\theta_\beta^*}[(\phi \otimes \phi)\exp(\beta^\top \psi)]}{\mathbb{E}_{\theta_\beta^*}[\exp(\beta^\top \psi)]} \tag{49}$$

$$= -\frac{\mathrm{Cov}_{\theta_\beta^*}[\phi \otimes \phi, \exp(\beta^\top \psi)]}{\mathbb{E}_{\theta_\beta^*}[\exp(\beta^\top \psi)]} \tag{50}$$

$$\stackrel{(a)}{=} -\frac{\mathrm{Cov}_{\theta_\beta^*}[\phi \otimes \phi, 1 + \beta^\top \psi + \mathcal{O}\left(\beta^2\right)]}{\mathbb{E}_{\theta_\beta^*}[1 + \mathcal{O}\left(\beta\right)]} \tag{51}$$

$$\stackrel{(b)}{=} -\mathrm{Cov}_{\theta_\beta^*}\left[\phi \otimes \phi, \ \beta^\top \psi\right] + \mathcal{O}\left(\beta^2\right), \tag{52}$$

where in (a) we used $\exp(\beta^\top \psi) = 1 + \beta^\top \psi + \mathcal{O}\left(\beta^2\right)$ and in (b) we used $\mathrm{Cov}[\cdot, 1] = 0$. The result again follows by taking expectations. $\qquad \square$

**Note:** Assuming that $\|\psi\|_1$ is small for most $z$ (as measured by $p_{\theta_\beta^*}$), the $\mathcal{O}\left(\beta^2\right)$ term is small as long as $\|\beta\|_\infty \ll 1$. This assumption on $\psi$ holds when $\mathbb{P}_{\theta_\beta^*}[\mathbb{S}] \approx 1$ (so that $\psi = 0$ most of the time).

*Proof of* (10). Recall that we are assuming $\beta_j = \beta_{\min}$ for all $j$, and that the $\neg\mathbb{S}_j$ are all disjoint. In this case, $-\beta^\top \psi$ is equal to $\beta_{\min}$ if a constraint is violated and 0 if no constraints are violated. We then have

$$\mathrm{Cov}_{\theta_\beta^*}\left[\phi \otimes \phi, \ -\beta^\top \psi\right] \tag{53}$$

$$= \beta_{\min} \mathrm{Cov}_{\theta_\beta^*}\left[\phi \otimes \phi, \ \mathbb{I}[\neg\mathbb{S}]\right] \tag{54}$$

$$= \beta_{\min}\mathbb{P}[\neg\mathbb{S}]\left( \mathbb{E}_{\theta_\beta^*}[\phi \otimes \phi \mid \neg\mathbb{S}] - \mathbb{E}_{\theta_\beta^*}[\phi \otimes \phi] \right) \tag{55}$$

$$= \beta_{\min}\mathbb{P}[\neg\mathbb{S}]\left( \mathbb{E}_{\theta_\beta^*}[\phi \otimes \phi \mid \neg\mathbb{S}] - \mathbb{P}[\neg\mathbb{S}]\mathbb{E}_{\theta_\beta^*}[\phi \otimes \phi \mid \neg\mathbb{S}] - \mathbb{P}[\mathbb{S}]\mathbb{E}_{\theta_\beta^*}[\phi \otimes \phi \mid \mathbb{S}] \right) \tag{56}$$

$$= \beta_{\min}\mathbb{P}[\mathbb{S}]\mathbb{P}[\neg\mathbb{S}]\left( \mathbb{E}_{\theta_\beta^*}[\phi \otimes \phi \mid \neg\mathbb{S}] - \mathbb{E}_{\theta_\beta^*}[\phi \otimes \phi \mid \mathbb{S}] \right), \tag{57}$$

as claimed. $\qquad \square$

## B.3 Optimizing $\beta$

*Proof of Proposition 3.3.* We can re-express $\mathbb{E}_{x,y\sim p^*}[-\log p(y \mid x)]$ as $\mathrm{KL}\left(p^* \| p\right) + H(p^*)$. Hence, in particular, $L(\theta, \beta) = \mathrm{KL}\left(p^* \| p_{\theta,\beta}\right) + H(p^*) \geq H(p^*)$,[2] with equality if and only if $p_{\theta,\beta} = p^*$. On the other hand, $p_{\theta^*,\infty} = p_{\theta^*} = p^*$ by assumption, so equality is attainable, and $(\theta^*, \infty)$ is a global optimum of $L$.

Note that the normalization constant $A(\beta)$ is important here, since if $p_{\theta,\beta}$ did not (sub-)normalize then the KL divergence would not necessarily be non-negative. $\qquad \square$

# C  Proofs for Section 4

*Proof of* (14). The acceptance rate is simply the expectation, over all $z \mid x$, of the acceptance probability for that particular $z$. This can clearly be written as

$$\sum_z p_{\theta,\mathbb{T}}(z \mid x) \exp\left(\beta^\top \psi(z,y)\right) \tag{58}$$

$$= \sum_z \mathbb{T}(z,y) \exp\left(\theta^\top \phi(x,z) - A_{\mathbb{T}}(\theta;x,y)\right) \exp\left(\beta^\top \psi(z,y)\right) \tag{59}$$

$$= \exp\left(-A_{\mathbb{T}}(\theta;x,y)\right) \sum_z \mathbb{T}(z,y) \exp\left(\theta^\top \phi(x,z) + \beta^\top \psi(z,y)\right) \tag{60}$$

$$= \exp\left(A(\theta,\beta;x,y) - A_{\mathbb{T}}(\theta;x,y)\right). \tag{61}$$

Since (14) is the multiplicative inverse of (61), the result follows. □

*Proof of* (15). By convexity of $A(\theta,\beta;x,y)$, we have

$$A(\theta,\beta;x,y) \tag{62}$$

$$\geq A(\tilde{\theta},\tilde{\beta};x,y) + (\theta-\tilde{\theta})^\top \nabla_\theta A(\tilde{\theta},\tilde{\beta};x,y) + (\beta-\tilde{\beta})^\top \nabla_\beta A(\tilde{\theta},\tilde{\beta};x,y) \tag{63}$$

$$= A(\tilde{\theta},\tilde{\beta};x,y) + (\theta-\tilde{\theta})^\top \mathbb{E}_{p_{\tilde{\theta},\tilde{\beta}}(\cdot|x,y)}[\phi(x,z)] + (\beta-\tilde{\beta})^\top \mathbb{E}_{p_{\tilde{\theta},\tilde{\beta}}(\cdot|x,y)}[\psi(z,y)] \tag{64}$$

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

[Supplementary Material 3 · appendix.pdf]

# A    Proofs for Section 2

*Proof of Proposition 2.5.* Let $\iota \overset{\text{def}}{=} \pi_1 \times \cdots \times \pi_k$. Since $\iota$ is injective by assumption, we have

$$\exp(A(\beta; z)) = \sum_{y \in \mathcal{Y}} \exp(\beta^\top \psi(z, y)) \tag{16}$$

$$= \sum_{p \in \iota(\mathcal{Y})} \exp(\beta^\top \psi(z, \iota^{-1}(p))) \tag{17}$$

$$= \sum_{p \in \iota(\mathcal{Y})} \exp\left(-\sum_{j=1}^k \beta_j \mathbb{I}[\pi_j(f(z)) \neq p_j]\right) \tag{18}$$

$$\leq \sum_{p \in \prod_j \mathcal{Y}_j} \exp\left(-\sum_{j=1}^k \beta_j \mathbb{I}[\pi_j(f(z)) \neq p_j]\right) \tag{19}$$

$$= \prod_{j=1}^k \sum_{p_j \in \mathcal{Y}_j} \exp\left(-\beta_j \mathbb{I}[\pi_j(f(z)) \neq p_j]\right) \tag{20}$$

$$= \prod_{j=1}^k \left(1 + (|\mathcal{Y}_j| - 1) \exp(-\beta_j)\right), \tag{21}$$

as was to be shown.    $\square$

# B    Proofs for Section 3

## B.1    Effect on loss

*Proof of Proposition 3.1.* Note that we have

$$L^* = \mathbb{E}_{p^*}[-\log p_{\theta^*}(y \mid x)] \tag{22}$$

$$= \mathbb{E}_{p^*}[-\log \mathbb{E}_{z \sim p_{\theta^*}}[\mathbb{S}(z, y)]] \tag{23}$$

$$\overset{(a)}{\geq} \mathbb{E}_{p^*}[-\log \mathbb{E}_{z \sim p_{\theta^*}}[\exp(\beta^\top \psi(z, y))]] \tag{24}$$

$$= \mathbb{E}_{p^*}[-\log \mathbb{E}_{z \sim p_{\theta^*}}[p_\beta(y \mid z) \exp(A(\beta))]] \tag{25}$$

$$= \mathbb{E}_{p^*}[-\log p_{\theta^*, \beta}(y \mid x) - A(\beta)] \tag{26}$$

$$= L(\theta^*, \beta) - A(\beta) \tag{27}$$

$$\overset{(b)}{\geq} L(\theta_\beta^*, \beta) - A(\beta). \tag{28}$$

Here (a) follows because $\mathbb{S}(z, y) \leq \exp(\beta^\top \psi(z, y))$, since the latter is non-negative and is 1 when $\mathbb{S}(z, y) = 1$; (b) follows because $\theta_\beta^*$ is the minimizer of $L(\cdot, \beta)$. Continuing:

$$L(\theta_\beta^*, \beta) - A(\beta) = \mathbb{E}_{p^*}[-\log \mathbb{E}_{z \sim p_{\theta_\beta^*}}[\exp(\beta^\top \psi(z, y))]] \tag{29}$$

$$\overset{(c)}{\geq} \mathbb{E}_{p^*}[-\log \mathbb{E}_{z \sim p_{\theta_\beta^*}}[\exp(-\beta_{\min}(1 - \mathbb{S}(z, y)))]] \tag{30}$$

$$= \mathbb{E}_{p^*}[-\log(p_{\theta_\beta^*}(y \mid x) + (1 - p_{\theta_\beta^*}(y \mid x)) \exp(-\beta_{\min}))] \tag{31}$$

$$= \mathbb{E}_{p^*}[-\log(1 - (1 - p_{\theta_\beta^*}(y \mid x))(1 - \exp(-\beta_{\min})))] \tag{32}$$

$$\overset{(d)}{\geq} \mathbb{E}_{p^*}[(1 - p_{\theta_\beta^*}(y \mid x))(1 - \exp(-\beta_{\min}))]. \tag{33}$$

Again, (c) follows because $\beta^\top \psi(z, y) \leq -\beta_{\min}(1 - \mathbb{S}(z, y))$, and (d) follows because $-\log(1 - x) \geq x$ for $x \leq 1$. Putting these together, we have $L^* \geq (1 - \exp(-\beta_{\min}))\mathbb{E}_{p^*}[1 - p_{\theta_\beta^*}(y \mid x)]$, which yields the desired result.    $\square$

*Proof of Lemma 3.2.* We will show a stronger result: any model and relaxation can be slightly modified to cause $\mathbb{E}_{p^*}[p_{\theta^*_\beta}(y \mid x)]$ to be zero, in a way that is demonstrated below (though the modified model will no longer be an exponential family).

Given any $\mathbb{S}_{1:k}$, construct a new point $z_0$ such that $\mathbb{S}_{1:k}(z_0, y) = 1$ for all $y$, and add a new constraint $\mathbb{S}_0(z, y) = [z \neq z_0]$. Then $\mathbb{S}(z_0, y) = 0$ for all $y$, so we never want to place mass on $z_0$ under the unrelaxed supervision. In addition, extend the model family to allow the single additional distribution $p'(z \mid x) = \mathbb{I}[z = z_0]$.

Now, suppose $\beta_{1:k} = \infty$ and $\beta_0 = \beta_{\min}$. Then, for any $\theta$, we have $L(\theta, \beta) = A(\beta) + L(\theta, \infty)$, since $p_\theta$ places no mass on $z_0$; therefore, $L(\theta, \beta) \geq A(\beta) + L^*$ for all $\theta$. On the other hand, we have $L(p', \beta) = A(\beta) + \beta_{\min}$. If $\beta_{\min} < L^*$, we will thus use $p'$ and shift all of the mass to $z_0$, thereby placing zero mass on the correct answer. $\square$

Note that the proof required constructing a "bad" $z_0$ that satisfied almost all the constraints for many values of $y$ at once. It seems straightforward to avoid this in practice, and so it would be interesting to find assumptions under which we obtain a better relative loss bound than Proposition 3.1.

## B.2   Amount of data needed to learn

For the next few derivations we will make extensive use of the relation $\log p_{\theta,\beta}(y \mid x) = A(\theta, \beta; x, y) - A(\theta; x)$, where $A(\theta, \beta; x, y) \stackrel{\text{def}}{=} \log \left( \sum_z \exp \left( \theta^\top \phi(x, z) + \beta^\top \psi(z, y) \right) \right)$. Note that the preceding definition is consistent with (13) since we assume throughout Section 3 that $\mathbb{T} \equiv 1$. We will also use the following properties of log-partition functions:

$$\nabla_\theta A(\theta, \beta; x, y) = \mathbb{E}_{z \sim p_{\theta,\beta}(\cdot \mid x, y)}[\phi(x, z)] \tag{34}$$

$$= \frac{\mathbb{E}_{z \sim p_\theta(\cdot \mid x)}[\phi(x, z) \exp(\beta^\top \psi(z, y))]}{\mathbb{E}_{z \sim p_\theta(\cdot \mid x)}[\exp(\beta^\top \psi(z, y))]}, \tag{35}$$

$$\nabla_\theta^2 A(\theta, \beta; x, y) = -(\nabla_\theta A)(\nabla_\theta A)^\top + \mathbb{E}_{z \sim p_{\theta,\beta}(\cdot \mid x, y)}[\phi(x, z) \otimes \phi(x, z)] \tag{36}$$

$$= -(\nabla_\theta A)(\nabla_\theta A)^\top + \frac{\mathbb{E}_{z \sim p_\theta(\cdot \mid x)}[(\phi(x, z) \otimes \phi(x, z)) \exp(\beta^\top \psi(z, y))]}{\mathbb{E}_{z \sim p_\theta(\cdot \mid x)}[\exp(\beta^\top \psi(z, y))]}. \tag{37}$$

Here we use $\nabla_\theta A$ as short-hand for $\nabla_\theta A(\theta, \beta; x, y)$. These $\nabla_\theta A$ terms will always cancel out in the sequel, so they can be safely ignored. (The cancellation occurs because we always end up subtracting two log-normalization constants, whose gradients must be equal by first-order optimality conditions.) Analogous properties to those above hold for $A(\theta; x)$:

$$\nabla_\theta A(\theta; x) = \mathbb{E}_{z \sim p_\theta(\cdot \mid x)}[\phi(x, z)], \tag{38}$$

$$\nabla_\theta^2 A(\theta; x) = -(\nabla_\theta A)(\nabla_\theta A)^\top + \mathbb{E}_{z \sim p_\theta(\cdot \mid x)}[\phi(x, z) \otimes \phi(x, z)]. \tag{39}$$

In this case, $\nabla_\theta A$ is short-hand for $\nabla_\theta A(\theta; x)$.

*Proof of* (8). We have

$$\mathcal{I}_\infty = \nabla_\theta^2[-\log p_{\theta^*, \infty}(y \mid x)] \tag{40}$$

$$= \nabla_\theta^2 [A(\theta^*; x) - A(\theta^*, \infty; x, y)] \tag{41}$$

$$= \mathbb{E}_{\theta^*}[\phi(x, z) \otimes \phi(x, z)] - \frac{\mathbb{E}_{\theta^*}[(\phi(x, z) \otimes \phi(x, z))\mathbb{S}(z, y)]}{\mathbb{E}_{\theta^*}[\mathbb{S}(z, y)]} \tag{42}$$

$$= \mathbb{E}_{\theta^*}[\phi(x, z) \otimes \phi(x, z)] - \mathbb{E}_{\theta^*}[\phi(x, z) \otimes \phi(x, z) \mid \mathbb{S}] \tag{43}$$

$$= (\mathbb{P}[\neg\mathbb{S}]\mathbb{E}_{\theta^*}[\phi \otimes \phi \mid \neg\mathbb{S}] + \mathbb{P}[\mathbb{S}]\mathbb{E}_{\theta^*}[\phi \otimes \phi \mid \mathbb{S}]) - \mathbb{E}_{\theta^*}[\phi \otimes \phi \mid \mathbb{S}] \tag{44}$$

$$= \mathbb{P}_{\theta^*}[\neg\mathbb{S}] (\mathbb{E}_{\theta^*}[\phi \otimes \phi \mid \neg\mathbb{S}] - \mathbb{E}_{\theta^*}[\phi \otimes \phi \mid \mathbb{S}]). \tag{45}$$

The result follows by taking expectations. $\square$

*Proof of* (9). We have

$$\mathcal{I}_\beta = \nabla_\theta^2[-\log p_{\theta_\beta^*,\beta}(y \mid x)] \tag{46}$$

$$= \nabla_\theta^2\left[A(\theta_\beta^*; x) - A(\theta_\beta^*, \beta; x, y)\right] \tag{47}$$

$$= \mathbb{E}_{\theta_\beta^*}[\phi(x,z) \otimes \phi(x,z)] - \frac{\mathbb{E}_{\theta_\beta^*}[(\phi(x,z) \otimes \phi(x,z))\exp(\beta^\top\psi)]}{\mathbb{E}_{\theta_\beta^*}[\exp(\beta^\top\psi)]} \tag{48}$$

$$= \frac{\mathbb{E}_{\theta_\beta^*}[\phi \otimes \phi]\mathbb{E}_{\theta_\beta^*}[\exp(\beta^\top\psi)] - \mathbb{E}_{\theta_\beta^*}[(\phi \otimes \phi)\exp(\beta^\top\psi)]}{\mathbb{E}_{\theta_\beta^*}[\exp(\beta^\top\psi)]} \tag{49}$$

$$= -\frac{\mathrm{Cov}_{\theta_\beta^*}[\phi \otimes \phi, \exp(\beta^\top\psi)]}{\mathbb{E}_{\theta_\beta^*}[\exp(\beta^\top\psi)]} \tag{50}$$

$$\overset{(a)}{=} -\frac{\mathrm{Cov}_{\theta_\beta^*}[\phi \otimes \phi, 1 + \beta^\top\psi + \mathcal{O}\left(\beta^2\right)]}{\mathbb{E}_{\theta_\beta^*}[1 + \mathcal{O}\left(\beta\right)]} \tag{51}$$

$$\overset{(b)}{=} -\mathrm{Cov}_{\theta_\beta^*}\left[\phi \otimes \phi, \ \beta^\top\psi\right] + \mathcal{O}\left(\beta^2\right), \tag{52}$$

where in (a) we used $\exp(\beta^\top\psi) = 1 + \beta^\top\psi + \mathcal{O}\left(\beta^2\right)$ and in (b) we used $\mathrm{Cov}[\cdot, 1] = 0$. The result again follows by taking expectations. $\qquad\square$

**Note:** Assuming that $\|\psi\|_1$ is small for most $z$ (as measured by $p_{\theta_\beta^*}$), the $\mathcal{O}\left(\beta^2\right)$ term is small as long as $\|\beta\|_\infty \ll 1$. This assumption on $\psi$ holds when $\mathbb{P}_{\theta_\beta^*}[\mathbb{S}] \approx 1$ (so that $\psi = 0$ most of the time).

*Proof of* (10). Recall that we are assuming $\beta_j = \beta_{\min}$ for all $j$, and that the $\neg\mathbb{S}_j$ are all disjoint. In this case, $-\beta^\top\psi$ is equal to $\beta_{\min}$ if a constraint is violated and $0$ if no constraints are violated. We then have

$$\mathrm{Cov}_{\theta_\beta^*}\left[\phi \otimes \phi, \ -\beta^\top\psi\right] \tag{53}$$

$$= \beta_{\min}\,\mathrm{Cov}_{\theta_\beta^*}\left[\phi \otimes \phi, \ \mathbb{I}[\neg\mathbb{S}]\right] \tag{54}$$

$$= \beta_{\min}\mathbb{P}[\neg\mathbb{S}]\left(\mathbb{E}_{\theta_\beta^*}[\phi \otimes \phi \mid \neg\mathbb{S}] - \mathbb{E}_{\theta_\beta^*}[\phi \otimes \phi]\right) \tag{55}$$

$$= \beta_{\min}\mathbb{P}[\neg\mathbb{S}]\left(\mathbb{E}_{\theta_\beta^*}[\phi \otimes \phi \mid \neg\mathbb{S}] - \mathbb{P}[\neg\mathbb{S}]\mathbb{E}_{\theta_\beta^*}[\phi \otimes \phi \mid \neg\mathbb{S}] - \mathbb{P}[\mathbb{S}]\mathbb{E}_{\theta_\beta^*}[\phi \otimes \phi \mid \mathbb{S}]\right) \tag{56}$$

$$= \beta_{\min}\mathbb{P}[\mathbb{S}]\mathbb{P}[\neg\mathbb{S}]\left(\mathbb{E}_{\theta_\beta^*}[\phi \otimes \phi \mid \neg\mathbb{S}] - \mathbb{E}_{\theta_\beta^*}[\phi \otimes \phi \mid \mathbb{S}]\right), \tag{57}$$

as claimed. $\qquad\square$

### B.3 Optimizing $\beta$

*Proof of Proposition 3.3.* We can re-express $\mathbb{E}_{x,y\sim p^*}[-\log p(y \mid x)]$ as $\mathrm{KL}\left(p^* \| p\right) + H(p^*)$. Hence, in particular, $L(\theta, \beta) = \mathrm{KL}\left(p^* \| p_{\theta,\beta}\right) + H(p^*) \geq H(p^*),$[2] with equality if and only if $p_{\theta,\beta} = p^*$. On the other hand, $p_{\theta^*,\infty} = p_{\theta^*} = p^*$ by assumption, so equality is attainable, and $(\theta^*, \infty)$ is a global optimum of $L$.

Note that the normalization constant $A(\beta)$ is important here, since if $p_{\theta,\beta}$ did not (sub-)normalize then the KL divergence would not necessarily be non-negative. $\qquad\square$

# C    Proofs for Section 4

*Proof of* (14). The acceptance rate is simply the expectation, over all $z \mid x$, of the acceptance probability for that particular $z$. This can clearly be written as

$$\sum_z p_{\theta,\mathbb{T}}(z \mid x) \exp\left(\beta^\top \psi(z,y)\right) \tag{58}$$

$$= \sum_z \mathbb{T}(z,y) \exp\left(\theta^\top \phi(x,z) - A_\mathbb{T}(\theta;x,y)\right) \exp\left(\beta^\top \psi(z,y)\right) \tag{59}$$

$$= \exp\left(-A_\mathbb{T}(\theta;x,y)\right) \sum_z \mathbb{T}(z,y) \exp\left(\theta^\top \phi(x,z) + \beta^\top \psi(z,y)\right) \tag{60}$$

$$= \exp\left(A(\theta,\beta;x,y) - A_\mathbb{T}(\theta;x,y)\right). \tag{61}$$

Since (14) is the multiplicative inverse of (61), the result follows. □

*Proof of* (15). By convexity of $A(\theta,\beta;x,y)$, we have

$$A(\theta,\beta;x,y) \tag{62}$$

$$\geq A(\tilde{\theta},\tilde{\beta};x,y) + (\theta - \tilde{\theta})^\top \nabla_\theta A(\tilde{\theta},\tilde{\beta};x,y) + (\beta - \tilde{\beta})^\top \nabla_\beta A(\tilde{\theta},\tilde{\beta};x,y) \tag{63}$$

$$= A(\tilde{\theta},\tilde{\beta};x,y) + (\theta - \tilde{\theta})^\top \mathbb{E}_{p_{\tilde{\theta},\tilde{\beta}}(\cdot|x,y)}[\phi(x,z)] + (\beta - \tilde{\beta})^\top \mathbb{E}_{p_{\tilde{\theta},\tilde{\beta}}(\cdot|x,y)}[\psi(z,y)] \tag{64}$$

$$= A(\tilde{\theta},\tilde{\beta};x,y) + (\theta - \tilde{\theta})^\top \tilde{\phi} + (\beta - \tilde{\beta})^\top \tilde{\psi}, \tag{65}$$

as was to be shown. □

## Footnotes

[2] Here we use the fact that $\mathrm{KL}\left(p \| q\right) \overset{\mathrm{def}}{=} \mathbb{E}_p[\log p - \log q]$ is non-negative as long as $p$ normalizes and $q$ sub-normalizes, which is true for $q = p_{\theta,\beta}$ by Proposition 2.5.