[Reviews · NeurIPS 2015]

Submitted by Assigned_Reviewer_1

# Summary of paper

The paper proposes a method for handling models with latent variables where the latent variables have a complex, deterministic relationship with the output variables. Computing marginal probability in such models is intractable because it requires summing over the latent variables. The proposed method forms an alternate probability distribution that decomposes a functional representation of the determinism, which essentially penalizes disagreement with the constraints.

The relaxation is theoretically analyzed, showing that the effect of the relaxation on loss is bounded by the inverse of the minimum value in the relaxation vector beta (which ranges from 0 to infinity), that the amount of data needed to learn increases by at most a similar ratio. The authors then provide analysis for the tradeoff between efficiency and these negative effects in setting the relaxation vector, which uses sampling to learn and adjust the relaxation parameter in a manner that guarantees tractable inference.

The method is tested on synthetic data for translation and parsing. The evaluation essentially only compares against different variants of the proposed approach, so it doesn't compare against any existing methods for similar problems. I don't see any discussion of this either. It seems the approaches described in the 2nd paragraph of the paper would be useful comparisons.

# Quality

This is a very good paper, with an innovative technical contribution, well evaluated and analyzed results, and good presentation. I'm actually left with a feeling the virtues of the proposed approach are too good. What are the weaknesses of this approach? What is the catch? This problem is related to the lack of comparison, in discussion or experiments, to other perhaps-less-principled approaches to problems with intractable hard constraints.

# Clarity

It is obvious to me that the authors took a lot of care in presenting what could otherwise be difficult, dense material. The writing is some of the best I have seen, with useful examples, readable, but technically precise exposition, and well thought out structure.

# Originality

The idea of relaxing hard constraints is not particularly new, but I think using the idea in this way is.

# Significance

The approach should apply to a variety of applications, and the technical ideas may extend to other problems.

Summary: I think this paper should certainly appear at NIPS. My only complaint is comparison to other approaches is lacking.

Submitted by Assigned_Reviewer_2

Summary: The authors propose an annealing framework for learning with relaxed constraints in structured prediction. They prove various consistency results of their model and how constraint satisfaction can be traded for inference tractability. Experimental evaluations show properties of their model, but are limited small data set size with small state/feature spaces.

Pros: - The authors propose a nice framework for trading off satisfying hard constraints against inference tractability in structured predictions by using one single annealing parameter \beta.

- There is a very thorough analysis on the theoretical properties of the model, such as model consistency and data sample requirements.

Cons: - The experiments focus on simulations of small sample size and small vocabulary. It is not clear if the method is scalable to more realistic dataset and vocabulary size. It is worrying that the authors mention that 10^4 samples are required per example for \beta=1 in one of the experiment settings.

- Although the authors have provided a nice framework for relaxing the constraints, it is not clear if this would perform better than simpler previous approaches such as constraint-driven learning [3] or enforcing posterior expectation agreement [7].

Summary: The authors propose an annealing framework for learning with relaxed constraints in structured prediction. They prove various consistency results of their model and how constraint satisfaction can be traded for inference tractability. Experimental evaluations show properties of their model, but are limited small data set size with small state/feature spaces.

Submitted by Assigned_Reviewer_3

This paper presents an approach to learn a model based on partial supervision when there are some deterministic constraints between output and latent variables; however, the inference between them is intractable.

Quality: the paper provides nice motivating examples and comprehensive discussions. Experiments results well support the theoretical discussions, but doesn't compare the proposed method to other alternative approaches.

Clarity: Overall, the paper is well-written, although some parts of it is unclear. Especially, the introduction section is not very motivated.

Originality: To my best knowledge, the proposed approach is new.

Significant: Learning with relaxed supervision is an important problem. The proposed approach may applies to many problems in computer vision an NLP.

this paper is interesting. However, some parts of the paper are not very clear. Please see some comments/questions below.

- I'm not sure if I understand what do the authors mean by " it is still intractable to incorporate the hard supervision constraint [S(z, y)=1]." To my understanding, in the problems that the authors are interested in, if z is given, estimate z(z, y) is easy. However, finding z to satisfy S(z,y) =1 is intractable. Is this what the authors mean?

- Maybe, the authors can move some examples in Section 2 to Section 1 to help readers understand the problem better.

- What is

z_i in Eq. (2)?

- Eq (5),(6): The authors should first discuss how q_\beta involve in p_{\theta, \beta} before given Eq (6)

- It seems to me an obvious alternative of the proposed approach is to use hidden CRF or latent SVM with an approximate inference (e.g., dual decomposition) that relaxes the constraints S(z,y)=1. Do the authors have consider this alternative?

- I don't really understand the argument in line 425-427.

Summary: Overall, I think this paper is interesting, and the scenario addressed by the paper may have many applications. The authors provide comprehensive theoretical discussions as well as implementation details about the proposed method.

Submitted by Assigned_Reviewer_4

Summary:

This paper considers a novel research direction within the broad research theme of "structured prediction with indirect supervision" that has wide range of applications in natural language processing. Specifically, authors' consider the problem of "learning from intractable supervision" in the form of hard global constraints between input, output, and latent variables. The inference problem (within the inner loop of learning) is intractable due to the hard constraints. Authors' study the trade-off between tractability of inference and quality of the learned models. The key idea is to define a relaxed supervision model (to the original model parametrized by \theta) with a parameter \beta resulting in a joint model parametrized by \theta and \beta; and jointly optimize (theta, \beta) to guarantee tractability of inference (rejection sampling) by making sure that the original model and the joint model are close to each other. Experiments are performed on two synthetic tasks to empirically verify the theoretical findings.

Pros:

- Novelty of the research questions! - Very nice formulation and formal analysis. - Experimental results verifying the theoretical findings.

Detailed Comments:

- Exposition of section 3 (especially "amount of data needed to learn") has a lot of room for improvement.

- Authors' should discuss the significance of their findings in the context of large body of work on structured prediction with indirect supervision (e.g., Posterior Regularization, Bayesian Measurements, and Generalized Expectation criteria).

- If possible, authors' should discuss how to extend the ideas to other inference procedures.
Summary: Summary:

This paper considers a novel research direction within the broad research theme of "structured prediction with indirect supervision" that has wide range of applications in natural language processing. Specifically, authors' consider the problem of "learning from intractable supervision" in the form of hard global constraints between input, output, and latent variables. The inference problem (within the inner loop of learning) is intractable due to the hard constraints. Authors' study the trade-off between tractability of inference and quality of the learned models. The key idea is to define a relaxed supervision model (to the original model parametrized by \theta) with a parameter \beta resulting in a joint model parametrized by \theta and \beta; and jointly optimize (theta, \beta) to guarantee tractability of inference (rejection sampling) by making sure that the original model and the joint model are close to each other. Experiments are performed on two synthetic tasks to empirically verify the theoretical findings.

Author Feedback
Author rebuttal: We would like to thank all of the reviewers for their helpful reviews; we appreciate the feedback and look forward to incorporating it into the final paper. We have no major disagreements with the points raised by reviewers, but will clarify a few minor points and respond to questions.

Assigned_Reviewer_1:
"What are the weaknesses of this approach? What is the catch?" -> One potential weakness is that the optimization problem (which is non-convex, as is typical for partially supervised problems) may become more difficult once we impose the tractability constraint. Another potential issue is that it is necessary to derive constraints that ensure tractability of inference, which for more complex inference procedures (e.g., MCMC) may end up being conservative. And of course, if we choose a very poor logical decomposition of the likelihood, we may get very little initial training signal. We do not think that any of these poses a major obstacle to the success of the proposed method, but hope that these examples are informative about what tradeoffs and assumptions exist within the framework.

Assigned_Reviewer_2:
"It is worrying that the authors mention that 10^4 samples are required per example for \beta=1 in one of the experiment settings." -> The point is to underscore the fact that a fixed value of beta may lead to poor performance, whereas optimizing beta subject to tractability constraints can lead to similar performance with substantially less computation.

Assigned_Reviewer_5:
"To my understanding, in the problems that the authors are interested in, if z is given, estimate z(z, y) is easy. However, finding z to satisfy S(z,y) =1 is intractable. Is this what the authors mean?" -> Yes, that is correct.

"What is z_i in Eq. (2)? " -> z_i is the i-th character of z. We will clarify in the text.

General comments / summary:
Some reviewers point out the lack of detailed experimental comparisons to other approaches, which we acknowledge and look forward to including in follow-up work. We feel the main strength of the paper is in the framework proposed, which to our knowledge is quite novel and able to handle new types of problems (such as semantic parsing and program induction) for which existing approaches are relatively unprincipled. In addition, some of the ideas (such as imposing tractability constraints or creating asymptotically consistent likelihood relaxations) can apply in other contexts, as well, which is why we are excited about disseminating these results to the NIPS audience.